# Seeing Structural Mechanisms of Optimized Piezoelectric and Thermoelectric Bulk Materials through Structural Defect Engineering

**DOI:** 10.3390/ma15020487

**Published:** 2022-01-09

**Authors:** Yang Zhang, Wanbo Qu, Guyang Peng, Chenglong Zhang, Ziyu Liu, Juncheng Liu, Shurong Li, Haijun Wu, Lingjie Meng, Lumei Gao

**Affiliations:** 1Instrumental Analysis Center of Xi’an Jiaotong University, Xi’an Jiaotong University, Xi’an 710049, China; menglingjie@xjtu.edu.cn (L.M.); lmgao@xjtu.edu.cn (L.G.); 2State Key Laboratory for Mechanical Behavior of Materials, Xi’an Jiaotong University, Xi’an 710049, China; snowar@stu.xjtu.edu.cn (W.Q.); 3121102069@stu.xjtu.edu.cn (G.P.); chenglongzhang@stu.xjtu.edu.cn (C.Z.); 3121302039@stu.xjtu.edu.cn (Z.L.); 3121102068@stu.xjtu.edu.cn (J.L.); 3121302056@stu.xjtu.edu.cn (S.L.); wuhaijunnavy@xjtu.edu.cn (H.W.)

**Keywords:** structural defects, STEM, piezoelectric, thermoelectric, polarization, phonon

## Abstract

Aberration-corrected scanning transmission electron microscopy (AC-STEM) has evolved into the most powerful characterization and manufacturing platform for all materials, especially functional materials with complex structural characteristics that respond dynamically to external fields. It has become possible to directly observe and tune all kinds of defects, including those at the crucial atomic scale. In-depth understanding and technically tailoring structural defects will be of great significance for revealing the structure-performance relation of existing high-property materials, as well as for foreseeing paths to the design of high-performance materials. Insights would be gained from piezoelectrics and thermoelectrics, two representative functional materials. A general strategy is highlighted for optimizing these functional materials’ properties, namely defect engineering at the atomic scale.

## 1. Introduction

For the majority of crystalline materials, the arrangement of a lattice is interrupted by various crystal defects, but such imperfections are important to the properties of materials. The properties of perfect crystalline materials would be only depended on their crystal structure and composition, which makes them hard to adjust. The probability of making defects beneficial allows us to customize functional attributes to the different combinations required by modern devices, effectively turning defects into advantages [1,2].

Crystal defects occur as points and lines in the form of a surface or distributed in the bulk, referred to as point, line, planar or bulk defects respectively. Here, atomic-scale defects refer to those with at least one dimension at the atomic scale, including all point defects, dislocations, grain/phase boundaries and interfaces of nanostructures, etc. Atomic-scale defects always induce static lattice distortion and influence thermal vibrations [3,4,5,6,7], especially under the action of an external thermal/stress/electric field.

Optimizing the properties of functional materials is a challenging task. Functional materials with various applications have different considerations for their properties. For thermoelectric materials, the semiconducting nature is of great significance, including characteristics like carrier concentration, mobility and band structure [8,9,10,11,12,13,14,15,16,17,18,19,20,21,22,23,24,25,26,27,28,29,30,31,32,33,34,35,36,37,38,39,40,41,42]. When it comes to ferroic materials, one should pay attention to the distribution of the order parameters, for example, polarization, spin, strain, and coupling [43,44,45,46,47,48,49,50,51,52,53,54,55,56,57,58,59,60,61]. Catalysts need sufficient accessible active sites and high activity [62,63,64,65,66,67,68,69,70,71,72,73,74,75,76,77]. Two-dimensional materials with special physical properties require accurate control of the electronic and chemical structures of their surfaces and edges [78,79,80,81,82,83,84,85,86,87]. Structural defects are ubiquitous for all functional materials, and totally control their properties [88,89,90,91,92,93,94,95].

Nanostructures have been widely recognized as the most universal method to improve the properties of various functional materials, but due to difficulties in quantifying structure and distribution, defects at the atomic level have usually been ignored [96,97,98]. Such defects are hardly visible by traditional methods such like X-ray diffraction and normal electron microscopy. Most research has been depended on assumptions instead of actual calculated statistics [71,99,100,101,102,103]. However, with the help of the new aberration-corrected (scanning) transmission electron microscopy (AC-(S)TEM), it is now possible to observe more defects at atomic scale with clarity than ever [4,104,105,106].

AC-(S)TEM has grown to be a powerful characterization platform for various types of materials [7,79,88,90,91,107,108,109,110,111,112]. Due to the feasibility of obtained multiple images and spectra at the same time, AC-(S)TEM could provide a variety of capabilities for accurate atomic imaging and mapping of chemical and electronic structures [88,90,91].These techniques are indispensable to observing and adjusting for all defects at the atomic scale, and are therefore invaluable for materials research [88,89,91,92].

On account of direct seeing of defects at the atomic scale through AC-STEM, we show the fundamental and significance of defect engineering in optimizing the overall properties of various types of functional materials [91,92].Here we present new insights gained from piezoelectric/ferroelectric and thermoelectric materials, two representative functional materials.

## 2. Piezoelectrics: Nano-Scale Coexistence of Phases with Gradual Polarization Rotation for High Piezoelectricity

Piezoelectrics, due to their ability of interconversion between electrical energy and mechanical strain, are widely applied in electro-mechanical devices. Environmental problems make it urgent to develop new lead-free piezoelectric materials with high performance [44,55,113,114,115,116,117,118,119,120,121,122,123,124]. The lack of basic comprehension of the mechanisms at local level has hindered development. Although great responses occurring near the phase boundaries are well-known [125], atomic-level understanding of this behavior is a great challenge. The synthesis of new materials is mainly through educated trial and error [119,126].

Recently, significant improvements have been achieved in the properties of lead-free (K,Na)NbO_3_, BaTiO_3_- and BiFeO_3_-based piezoceramic material via composition tuning, in order to tune the phase boundaries into the desired temperature range as well as to improve the temperature stability [44,114,127]. The primary (K,Na)NbO_3_ and BaTiO_3_ successively underwent phase transition from the paraelectric cubic (*C*) phase to the ferroelectric tetragonal (*T*), orthorhombic (*O*) and rhombohedral (*R*) phases [113,116]. A phase boundary engineering method was employed to converge three phase transitions into an *R-T* phase transition for KNN-based system [44,114], and a quadruple critical point (QCP) for BaTiO_3_–based system [127]. The successful employment of such phase boundary engineering is actually guided by the informed understanding and expectation of the necessary structural imperfection.

### 2.1. (K,Na)NbO_3_-Based Piezoelectrics with Constructed R-T Phase Boundary

The *R-T* phase boundary shows wide phase coexistence and largely enhances the temperature stability of the piezoelectric parameters, as shown in Figure 1a–c. In system (1−*x*)(K*_z_*Na_1−*z*_)(Nb_1−*w*_Sb*_w_*)O_3−_*x*Bi_0.5_(Na_1−*y*_K*_y_*)_0.5_HfO_3_ (KNNS-BNKH) piezoceramics, a giant piezoelectric coefficient *d*_33_ of ~525 pC/N has been achieved, as well as great temperature stability with an effective piezoelectric coefficient *d*_33_* varying (10% in the temperature range of 27–80 °C [58,114]. Further enhancement (*d*_33_ ~550 pC/N) with high Curie temperature *Tc* (237 °C) has also realized in the (1−*x−y*)K_1−*w*_Na*_w_*Nb_1−*z*_Sb*_z_*O_3−_*x*BiFeO_3_-*y*Bi_0.5_Na_0.5_ZrO_3_ (KN*_w_*NS*_z_*−*x*BF_−_*y*BNZ) system via modifying the *R-T* phase boundary of KNN via optimizing *x*, *y*, *z*, *w* [44]. The *d*_33_ and *T*_C_ properties of KNN materials are far superior to all lead-free piezoelectric ceramics that have been reported previously, as shown in Figure 1b,c, and are furthermore nearly on par with the best (Pb,Zr)TiO_3_-based ceramics as a giant breakthrough, as shown in Figure 1c [44,114].

Microstructures, such as ferroelectric domains, could significantly control piezoelectric performance. An in-depth understanding of the microstructure, especially the local structure within the domain, is a prerequisite for revealing the structure-property relationship. Like other high-performance piezoelectric ceramics with phase boundary compositions, these materials possess hierarchical nanodomains [115,128,129,130,131]. In this case, fine nanodomains with widths of 10–30 nm and lengths of 100–300 nm are lined in the submicron domain (oriented along <100> directions), parallel to each other along <110> directions, as shown in Figure 1d. 

Convergent-beam electron diffraction (CBED) was applied in detection of symmetry information in nano-scale domains such as: the coexistence of *T* and *R* nanotwins [114]. The size of CBED probe is a few nanometers, which can identify symmetry information within such nanodomains [115], which is hard to achieve that through X-ray and Neutron diffraction because of the limitation of spatial resolution. The distortions of the low-symmetry *T* and *R* phases of KNN could be considered as elongations of their parent cubic unit cell along an edge ({001} for *T*), or along a body diagonal ({111} for *R*). The spontaneous polarization (Ps) directions, which import characteristic symmetry elements, could be identified via CBED. The *T* (*P4mm*) lattice is shown with a 4-fold rotation axis along <001> as well as with mirror planes along {100}/{110}, while the *R* (*R3c*) lattice presents with a 3-fold rotation axis as well as glide plane along <111>. The CBED patterns shown in Figure 1(e1,e2) reflect the local symmetry of *T* and *R* nanophases coexisting in the nanodomain, where they intersect and finally form the hierarchical domain structure. Moreover, the electron diffraction patterns shown in Figure 1(f1,f2) disclose weak superlattice spots which should not exist in the well-known KNN structure. The superlattice spots reflect local ordering existing inside the nanodomains. 

Although the success of local symmetry identification via the CBED technique was achieved, it only gives reciprocal-space information, so it is difficult to show atomic displacements or polarization in real space. Then aberration-corrected STEM was employed to directly observe the atom displacement, which could quantitatively reflect the polarization state and symmetry in per unit cell [132]. A STEM high angle angular dark field (HAADF) generates Z-contrast imaging, thus it is a useful structure image mode, especially at the atomic scale [108,109,110,111].

STEM HAADF was employed to identify the coordinates of a perovskite ABO_3_ lattice (A for K/Na and its substitutions while B for Nb and its substitutions) precisely, and to locate the local symmetry via the displacement vector of atom B. Figure 2b is a STEM HAADF lattice image gained on a domain boundary. Thanks to the Z-contrast difference, the A and B sites of the perovskite ABO_3_ lattice could be clearly recognized. A peak finder strategy was applied to identify the coordinates of atom columns precisely [43,44,133,134,135], as shown in Figure 2b. Figure 2(c1,c2) show enlarged versions from Figure 2b, indicating that the displacement vectors of the centers relative to the corners are variable, some are arranged along <100> while others on <110>, and they are in keeping with the schematics of the *T* and *R* unit cells seen in Figure 2(a1,a2). The local symmetry inside the nanodomains is a reflection of the *T* and *R* coexistence phases. 

Then quantitative measurement of atom displacements (i.e., polarization) through HAADF imaging was performed, which is always for heavy element characterization, and annular bright field (ABF) imaging was performed to identify light elements. Using these methods, local symmetries inside the domains, their relative concentrations, and polarization rotation between domains can be characterized. Figure 2(d1,d2) present STEM HAADF and ABF images, light elements, e.g., oxygen could be observed. The relative displacements of the corner between Nb atoms and O atoms could be mapped after locating all types of atom positions, as shown in Figure 2e, which indicates that the polarization rotates continuously between *R* and *T* phases. 

### 2.2. BaTiO_3_-Based Piezoelectrics with Constructed Wild R-O-T Phase Boundary Region

When it comes to BaTiO_3_–based piezoelectrics, a special engineering method was employed based on phase boundaries, utilizing a QCP to achieve first, the highest piezoelectric effect ever reported in any lead-free piezoelectric materials and even higher than the commercialized PZT, and second, a record broad temperature/composition plateau with high *d_33_* in lead-free BaTiO_3_ piezoceramics, as shown in Figure 3a,b. The successful employment of such phase boundary engineering is actually guided by the informed understanding and expectation of the necessary structural imperfection. 

Figure 3c shows STEM ABF as a quantitative analysis of the atomic displacements (i.e., polarization) in a BaTiO_3_-based system. STEM ABF images are useful when observing light elements, because they has a poor Z-dependence compared with HAADF [79,88], so they could be used to identify light oxygen positions. For ferroelectric BaTiO_3_, the spontaneous polarization (*P*_S_) shown in the inset of Figure 3c comes from the electric dipoles from relative displacements between negative (O^2−^) and positive (Ba^2+^ and Ti^4+^) ion centers, and the relative displacement of the central Ti^4+^ cation with respect to its two nearest O^2−^ neighbor (*δ*_Ti-O_) centers reflects the local polarization state, and therefore the symmetry, as shown in Figure 3c. The local *P*_S_ could be roughly calculated by a linear relation to *δ*_Ti-O_ (*P*s = *kδ*_Ti-O_, where *k* is a constant, ~1894 (μC cm^−2^) nm^−1^ for BaTiO_3_) [136]. The visualization of the 2D *δ*_Ti-O_ (polarization) vectors marked with polarization vectors for *T*, *O* and *R* phases is shown in Figure 3c. This can be clearly observed the coexistence of *T*, *O* and *R* nanoregions and the continuous polarization rotation between these nanoregions, which is not homogeneous, as shown in Figure 3d.

It necessary to interpret the origin of high piezoelectricity from a theoretical view. According to density functional theory (DFT) calculations, the addition of Sn or Ca could change the order of the stability of different phases, and the three ferroelectric phases (*R*, *T*, and *O*) possess almost zero energy difference. The quadruple critical point (QCP) composition (*C*+*T*+*O*+*R*) has almost isotropic free energy, which is independent of the direction of polarization, as shown in Figure 3g. The *T*+*O*+*R* three-phase coexistence components show stable *O* <110> and *R* <111> states as well as metastable *T* <100> states, compared to pure BaTiO_3_, polarization anisotropy is greatly reduced. In addition, phase-field modeling was employed to simulate the *T*+*O*+*R* multiphase coexistence state. As shown in Figure 3e, the *T*, *O*, and *R* phase states coexist in the nanodomain and permeate each other in random distribution. Therefore, there are various paths of polarization conversion between phases. Figure 3f shows multiphase coexists polarization projection on the {110} plane. The prevalent polarization rotation between *T*, *O*, and *R* phases is predicted, which is in good agreement with STEM results in Figure 3c. In summary, the coexistence of *T*+*O*+*R* multiphase with low free energy/polarization anisotropy, weak energy barrier, and multiple polarization rotation possibilities leads to a high piezoelectric coefficient.

According to the atomic polarization mappings and theoretical calculations, the physical origin of good piezoelectric properties at the phase transition region is the phase coexistence (*T+O+R*) inside hierarchical nanodomains, and the gradual polarization rotation is a bridge between different phases. This static polarization state may simulate dynamic polarization changes under external stimuli (heat or electric field). This kind of origin is common in piezoelectric materials with phase boundary components [44,114,115,128,129]. It is extremely important to understand the roles of such atomic-scale coexisted phases and the gradual polarization rotation between them on high piezoelectricity at phase boundaries, which is the base for further designing new materials with higher performance.

### 2.3. BiFeO_3_-Based Piezoelectrics with Strain-Driven R-T Phase Boundary 

The other promising lead-free piezoelectric materials are BiFeO_3_-based, presenting much higher T_C_ (~500 °C), compared with KNN and BaTiO_3_ [137], however it is challenging to achieve good ferroelectric/piezoelectric properties of BFO, because of its high leakage current. High quality BFO thin films have been recently achieved through strain engineering [138,139,140,141,142]. They are promising for high-density memory and spintronic devices. One of the characteristic achievements in BFO thin films was to construct a strain-driven phase boundary [141], which completely differs from the traditional chemical approaches. Zeches et al. demonstrated how to employ epitaxial strain to drive the formation of a phase coexisted boundary and thus produce a giant piezoelectric response in lead-free ferroelectric thin films. The phase coexistence of the tetragonal (T) nanoscale phase within the parent R-phase matrix is beneficial for the ferroelectric/piezoelectric performance. Both atomic force microscope (AFM) and TEM images show strip nanodomains which might be attributed to the coexisted phases, as shown in Figure 4a,b. The phase coexistence was directly seen via aberration-corrected STEM. The atomically-resolved STEM HAADF image in Figure 4c–e clearly differentiated the local R and T phases. Figure 4f gives the relative fractions of these two coexisting phases with respect to the film thickness. The phase coexistence could happen when the film thickness >50 nm. Moreover, the substrates with different lattice mismatches show different local structures in BFO film. The aberration-corrected STEM observation can support the assumption of a strain-driven phase boundary in thin film.

## 3. Perovskite Thermoelectric Oxides: The Bridge between Piezoelectrics/Ferroelectrics and Thermoelectrics

As discussed above, good piezoelectrics are always insulators when minimized electrical conduction is desired. A good thermal material has high electron conductivity, like a metal. When ferroelectrics become highly conductive, they destabilize the long-range dipolar ordering necessary for ferroelectricity. There were few clues in classic textbooks that hinted that telluride materials had the kind of crystal symmetries that were consistent with ferroelectricity, but because they had such high electrical conductivity, they could not switch polarization, a requirement of true ferroelectricity. However, weak piezoelectric/ferroelectric or even paraelectric oxides in the unusual condition where the concentration of electronic carriers is close to a metal–insulator transition have properties of interest for oxide-based thermoelectric applications. The typical example, doped SrTiO_3_, is paraelectric in bulk, while it could be ferroelectric in films under certain strained conditions [60,143,144,145]. The heavily reduced, nonstoichiometric n-type perovskite SrTiO_3−δ_ shows metallic-like conductivity [146,147,148,149]. 

SrTiO_3_-based thermoelectric oxides have attracted considerable attention due to their thermally stable features, compared with conventional semiconductor-based thermoelectric materials. With donor-doping with a higher valence ion on the Sr site, like La-doped SrTiO_3_, the overall thermoelectric performance of SrTiO_3_ has been improved remarkably, making this material system promising for high-temperature usage [146,147,148,149]. To compensate for the extra positive charge from the substitution of Sr^2+^ by La^3+^, A-site vacancies might form according to the general formula Sr_1−3*x*/2_La*_x_*TiO_3_. Lu et al. investigated the structure and thermoelectric properties of Sr_1−3*x*/2_La*_x_*TiO_3_ ceramics with different content of La dopants. It as shown that the thermoelectric properties, especially the electrical transport, are highly sensitive to the content of La, as shown in Figure 5a,b. Advanced electron microscopies, including aberration-corrected STEM, were employed to reveal the structural origin of this phenomenon. The samples with 0.10 ≤ *x* < 0.30 presented as overall cubic structure with superstructure (Figure 5(c1)); the samples with 0.30 ≤ *x* < 0.50 exhibited additional short-range A-site vacancy ordering (Figure 5(c2)); and the samples with *x* ≥ 0.50 were orthorhombic with a tilt system and long-range vacancy ordering (Figure 5(c3)). TEM images of Sr_1−3*x*/2_La*_x_*TiO_3_ with *x* = 0.50 along <110> revealed antiphase boundaries associated with antiphase rotations of the O-octahedra. For the sample with *x* = 0.63, it showed ferroelastic domains with orthorhombic distortion. The key feature of vacancy ordering can be directly seen via aberration-corrected STEM. As shown in Figure 5(e1,e2,f1,f2), two types of domains with normal perovskite and layered structures exist. The layered structure was formed due to cation vacancy ordering. 

## 4. Thermoelectrics: Structural Defect Engineering for Carrier and Phonon Transport

Thermoelectricity, which enables direct conversion between electrical and thermal energy, promises to harvest electric energy from waste heat sources and from the overheating of solid-state refrigeration electronics. Sizes of various structural defects have a strong relationship with their electrical and thermal transport properties [17,20,89,150,151,152,153,154,155,156,157,158]. The electrical transport characteristics (Seebeck coefficient *α* and electrical conductivity *σ*) of thermoelectrics are affected by the electronic band structure, interactions of carriers with natural lattice vibration, nanostructure and point defects, which can be expressed in Boltzmann transport equations within the approximate relaxation times [14,18,89,152,159].
(1)$$\sigma =\frac{e^{2}}{m_{I}^{*}}\frac{{\left(2m_{d}^{*}k_{B}T\right)}^{3/2}}{3\pi^{2}\hbar^{3}}\langle \tau \left(\varepsilon \right)\rangle$$
(2)$$\alpha =\frac{k_{B}}{e}\frac{\langle \tau \left(\varepsilon \right)(\varepsilon -\varepsilon_{F})\rangle }{\langle \tau \left(\varepsilon \right)\rangle }$$
(3)$$\tau {\left(\varepsilon \right)}^{-1}=\tau_{AC}{\left(\varepsilon \right)}^{-1}+\tau_{PD}{\left(\varepsilon \right)}^{-1}+\tau_{P}{\left(\varepsilon \right)}^{-1}$$
where $m_{I}^{*}$ = (1/$m_{L}^{*}\;$+ 2/$m_{T}^{*}$)^−1^ is the inert effective mass ($m_{T}^{*}\;$and $m_{L}^{*}$ are transverse and longitudinal mass), $m_{d}^{*}$ =$\;N_{V}^{2/3}m_{d}^{*}$ is the density of state (DOS) effective mass ($N_{V}\;$is the band degeneracy and $m_{d}^{*}$ the effective mass of single valley), $k_{B}$, ℏ and *e* are Boltzmann constant, Plank’s reduced constant and electron charge. ε = E/$k_{B}$T, ε_F_ = E_F_/$k_{B}$T are reduced energy and reduced Fermi energy, $\tau_{AC}$, $\tau_{PD}$ and $\tau_{P}$ are the relaxation time due to acoustic phonon scattering, point defect scattering and precipitate scattering, respectively [14,18].

When it comes to thermal transport, the lattice thermal conductivity can be expressed using Callaway’s model [160]:(4)$$\kappa_{lat}=\frac{k_{B}}{2\pi^{2}\upsilon }{\left(\frac{k_{B}T}{\hbar }\right)}^{3}\int_{0}^{\theta_{D}/T}\tau_{c}\frac{x^{4}e^{x}}{{\left(e^{x}-1\right)}^{2}}dx$$
(5)$$\tau_{c}^{-1}=\tau_{U}^{-1}+\tau_{N}^{-1}+\tau_{S}^{-1}+\tau_{D}^{-1}+\tau_{PD}^{-1}+\tau_{P}^{-1}+\tau_{B}^{-1}+\ldots$$
where υ is average sound velocity, *θ_D_* is Debye temperature, and x is defined as ℏω = *k_B_T*. The frequency (ω)-dependent τ_c_ is the integral of the relaxation time of different scattering processes, including the Umklapp process ($\tau_{U}$), normal processes ($\tau_{N}$), dislocations ($\tau_{D}$), strain ($\tau_{S}$), point defects due to solid solution ($\tau_{PD}$), nanoscale precipitates ($\tau_{P}$), and grain/phase boundaries ($\tau_{B}$), all these processes (with the exception of Umklapp and normal processes, which lie on the intrinsic lattice and bonding characteristics) are closely related to the structural imperfections at microscale, nanoscale and atomic-scale [89,150,161]. Gorai et al. contributed a great review on the computationally guided discovery of thermoelectric materials [162].

Here, representative examples are presented to illustrate various structural defects and their relationship with electrical and thermal transport performance. The defects incorporate submicron grains with compact low angle grain boundaries (Figure 6a–c), segregated precipitate at the grain boundary (Figure 6d–f), stacking faults (Figure 6g,h), platelet-like nanostructures in the PbTe-based matrix (Figure 7a,b), precipitates with different shapes, e.g., laminate (Figure 7c) and cubical (Figure 7d–h), as well as the resulting strain fields around them.

Submicron grains come from the spark plasma sintering method [151] or the hot-pressing sintering method [163], an indispensable synthesis strategy in thermoelectrics because the mechanical properties of pristine ingot are usually too weak to be used [151,163]. The grain size could be even lowered into nanoscale by solvothermal method combined with spark plasma sintering [164]. Submicron and nanoscale grains produce dense low-angle grain boundaries with arrays of atomic-scale dislocations that serve as significant phonon scattering centers [14], in particular, for phonons with long waves [151,163]. Under some circumstances, the segregation of the precipitated phase occurs at grain boundaries. As shown in Figure 6d–f, at the grain boundary, there are high density nano-meter precipitates segregated at triple junctions. The HRSTEM HAADF image in Figure 6f concentrates on a Bi precipitate at the grain boundary as well as on a Bi-rich precipitate inside the grain. Its strain analysis through geometric phase analysis (GPA) indicates that high strain centers are arranged between the Bi precipitates and the matrix, corresponding to the interface dislocation core. In addition, Bi nanoprecipitates can not only release the strain between the mismatched grains, but can also further promote charge redistribution as additional carriers, which are parallel to the result of modulated doping [21,22,165,166].

In addition to the structural defects on the grain boundaries, other type of plane defects exist, like stacking faults. Figure 6g shows a large amount of stacking faults. In the HRSTEM HAADF the fine structure of a stacking fault clearly emerges, as shown in Figure 6h. Such atomic-scale 2D planes of crystal mismatches densely pack together to form a 3D strain network, which is an effective scattering source for phonons with short to medium wavelengths. Furthermore, another type of planar defect, platelet-like precipitates with one/two atom-layer thickness, are characteristic nanostructures in lead chalcogenide (PbQ, Q = Te, Se and S) thermoelectrics [13,14,164,168,169]. These nanostructures are thought to be inherent, which comes from the inevitable evaporation of lead during its formation process. As shown in Figure 7a,b, platelet-like nanostructures are perpendicular or parallel with each other, keeping within two of three possible {100} directions. The GPA strain analysis in the inset of Figure 7b indicates an anisotropic strain distribution, compared with normal spherical or ellipsoidal ones [152].

Nanocrystalline precipitation is a characteristic feature of thermoelectric materials with nano-structures. For example, Okhay et al. contributed a good review about impact of graphene or reduced graphene oxide on performance of thermoelectric composites, including chalcogenides, skutterudites, and metal oxides [170,171]. Nanostructures have been the main structural defects for scattering phonons with short to middle wavelengths, which depends on the size and morphology of precipitation. Regular shapes were found that layer Pb, Bi poor phase in a SnTe system [172], and the rod-like Bi rich phase in a Mg_3_Sb_2_ system [167], as shown in Figure 7c. The lattice of a precipitate is very similar to that of the matrix, and there is no lattice mismatch, so the GPA strain analysis indicates a homogenous strain distribution. Nanoprecipitates with atomic-scale coherent interfaces can scatter phonons efficiently without causing too much disturbance to carrier transport. As shown in Figure 7e,f,(g1,g2), the Cu_2_Te precipitate which was identified by the EDS method was well-faceted. The electron diffraction patterns show centrosymmetric peak splitting, which reflects the epitaxial orientation relation between the layered Cu_2_Te (space group: P6/mmm) and cubic PbTe structures, as shown in Figure 7e. These phase boundaries can effectively scatter phonons without influencing carrier transport due to small lattice mismatch between two phases. The strain distribution around the Cu_2_Te precipitated phase was obtained by GPA [89]. According to the strain analysis in Figure 7f, the phase boundary has dislocation cores, and the layered Cu_2_Te precipitated phase presents periodical strain distribution. Figure 7h shows STEM HAADF/ABF lattice images of the Cu_2_Te precipitated phase with a four-layer structure. The insert shows the enlarged image around the superimposed fault, and high strain can be observed in Figure 7f. These structural defects may provide additional phonon scattering sources.

The nanostructured approach has been commonly recognized as the most common method for improving thermoelectric performance, but the defects at the atomic scale may play more significant roles on carrier and phonon transportation. With the new generation of AC-STEM, it is a great chance to make the direct visualization of atomic-scale defects possible. One of the most recent results was the direct observation of inherent Pb vacancies and extrinsic Cu interstitials so as to reveal the magic function of Cu on the synergistic majorization of phonon and carrier transport in traditional PbTe, as shown in Figure 8.

The doped Cu atoms in an intrinsic Pb vacancy could enhance the carrier mobility, as shown in Figure 8a,b while reducing the thermal conductivity of the lattice, as shown in Figure 8c, via scattering all-wavelength phonons from forming precipitates, clusters and interstitials. As shown in Figure 8a, the carrier mobility increases firstly, and then decreases. The substituted Cu atom in a Pb site shows a+1 valence state, which provides one less charge to the matrix (Pb^2+^), resulting in the reduction of the carrier concentration. On the other hand, interstitial copper atoms act as impurity dopants in the matrix, providing additional charge to add the carrier concentration, so the tendency of carrier concentration to change with the increase of Cu_2_Te content is the result of these two competitive effects. This large enhancement of carrier mobility is remarkable due to the occupancy of Pb vacancies by external dopants and has not been reported in any thermoelectric bulk materials with nanostructure. To comprehend the abnormal behavior of Cu in PbTe, it is necessary to focus on the formation energies of any possible defects (vacancies, antisites, interstitials, and Cu-filled inherent vacancies) in PbTe-Cu_2_Te. The formation energies of Cu-related defects can reveal the influence of copper on the electric transport performance. After Cu additions, Cu interstitials (Cu_i_^1+^) donate electrons and reach up to a higher-level of Fermi energy, leading to positive effects in *n*-type conductivity. The formation energy of Cu interstitials is higher than that of Cu-filled Pb vacancies. Cu will fill the Pb vacancy as an acceptor until the Pb vacancy becomes unavailable. Hence, the carrier concentration decreases at first, and then increases with increasing Cu fractions [4].

Based on the anomalous variation in electrical properties, Figure 8d gives a schematic figure showing how Cu atoms present in the matrix with increasing Cu fraction. At first, a small number of Cu atoms filled the inherent Pb vacancy in PbTe, which eliminated Pb vacancies and then diminished the carrier scattering and effectively improved the carrier mobility. With further increase of the Cu content, the excess Cu atoms were forced into the interstitial sites, forming isolated Cu interstitial arrays first, then Cu interstitial clusters, and finally Cu-rich precipitates and even Cu_2_Te precipitates. These layered structures could effectively scatter phonons across various length scales and lead to extremely low lattice thermal conductivity, as shown in Figure 8c [4,173]. In order to estimate whether point defects exist, AC-STEM was utilized to observe atomic-scale Pb vacancies and Cu interstitials. Interstitial arrays and clusters of copper could be seen in the magnified images, as shown in Figure 8e–g. In addition, Cu interstitials can cause local lattice distortion as well as local strain. The GPA strain analysis in Figure 8h exhibits the strain network caused by the Cu interstitials [4].

It’s worth noting that the ab initio molecular dynamics (AIMD) calculations show a farther synergy: the copper atom vibrates near the lead vacancy with a maximum displacement of about 3.4 Å. Cu atoms in the region shown in Figure 8i exhibit highly anisotropic vibrations along {110} direction. In addition, it obviously interferes with the movement of Pb atoms around it. Cu atoms cause local lattice disorder, which plays an important role in the scattering of phonons with high frequency at high temperatures [4,43,174].

In addition to vacancies and interstitials, another important type of point defects, substitutions, could also been utilized to optimize the electrical and thermal transport. In thermoelectrics, the well-employed band structure engineering strategies to boost the Seebeck coefficient, e.g., band alignment and band gap enlargement, are highly related with substitutions, like Sr doped in PbTe [151,152], Mg doped in PbTe [13], and Se doped in PbTe [10]. For SnSe, Te alloying as a substitution could increase the crystal symmetry, optimize the bond structure, change the band shape, and thus enhance the electrical transport properties. Meanwhile, the substitutions could play as phonon scattering centers and contribute to lower the thermal conductivity [33]. AC-STEM revealed the substitution of Te dopants at Se sites and its effluence at atom bonds. Figure 9a is an atomic resolution STEM HAADF image along {100} region axis (a axis), showing a dumbbell-shaped arrangement of atoms. Each atom column is not round, but rather slightly elongated and mismatched because half of the Sn and Se atoms overlapped. The two columns of the dumbbell were actually of equal intensity. To better view the substitution of Te on Se or Na on Sn sites, it turns to the b or c axis, because the Sn and Se columns could be nicely separated along these axes. Figure 9d is an atomically resolved STEM HAADF image along {001} region axis (c axis). It is clear that Sn and Se atoms are well distinguished by their apparent intensity differences. The respective electron diffraction patterns in Figure 9e show the reflection of the superlattice due to multiple cycles in arrangement of the atoms, as opposed to the case along the a axis (Figure 9b). 

To assess Te substitutions and bonds, a quantitative analysis of Figure 9d was performed on atom positions and intensity, with a peaking methodology. Here, the atom columns were divided in 4 sets, Sn1, Sn2, Se3 and Se4. Figure 9g shows the intensity mapping of Sn1 atom columns. The uneven intensity distribution indicates the presence of Te substitution (abnormal bright column) at the Sn site. The bond lengths of Sn-Sn and Sn-Se can be acquired from the determined atomic positions. Figure 9h,i calculate the lattice parameters of Sn1 atoms columns, associated with the bond lengths of Sn-Sn along X (020) and Y (400) directions. Once the positions of all the atomic columns were determined, the bond lengths of Sn-Se could be reflected by their X and Y projections, as shown in Figure 9h–j. All of these maps reflect a feature simultaneously: uneven contrast and slight deviations. Intensity and bond length difference resulted from Te replacing Se sites is tiny, because few of the substituted atoms are buried in the thicker matrix atom column (~a few dozen atoms). The substitution of Te on Se causes changes in local bond lengths and local strain fields, which have great influence on electric and thermal transmission [33].

## 5. Conclusions and Prospects

AC-STEM/TEM could realize various functions for accurate atom imaging, chemical mapping, electronical configuration, etc, because multiple images and spectra can be obtained simultaneously. These structural features are closely related to the properties of materials and are of great value to materials research. 

For piezoelectric materials, the quantitative atom displacement calculation from STEM Z-contrast images was developed into a common method for characterizing the local polarization configuration, which is significant to unveil the structural and physical issues behind high performance at phase boundaries. The increase of theresolution of the AC-STEM will implement an effective application in ferroelectricity: 3D polarization mapping through fine control of optical depth slices [175,176]. It is significant to fully understand the responses to local polarization in practical situations, and not just in two-dimensional projections. 

For thermoelectric materials, structural defects at various scales have been recognized as the main parameters for optimizing carrier and phonon transport characteristics. The quantification of atom defects has most often been ignored in conventional methods, because of its difficulty. Using AC-STEM, we can see that, in contrast to widely accepted nanoscale structures, intrinsic and extrinsic defects at the atomic scale may become dominant. With the exploitation of these new-generation thermoelectric materials, for example, SnSe [8,9,12], the intrinsic defects of these materials at the atomic scale have attracted widespread attention. Atomic scale defects are always present in thermodynamic states. Local disorders and related anomalies of local lattice thermal vibrations are widespread in thermoelectric materials, particularly at extreme temperatures. The key to increasing thermoelectric efficiency is to manipulate the dynamics of atoms and their defects in lattices. Atomic-scale point defect engineering will become a new strategy to improve both electrical and thermal properties of thermoelectric materials. Despite the significant progress achieved, it has been severely hampered by lack of any direct micro information about these atomic scale defects, particularly their dynamic vibrational properties. With the advent of a new generation of AC-STEM, this has become possible, and it will further lead to improved performance.

## Figures and Tables

**Figure 1 materials-15-00487-f001:**
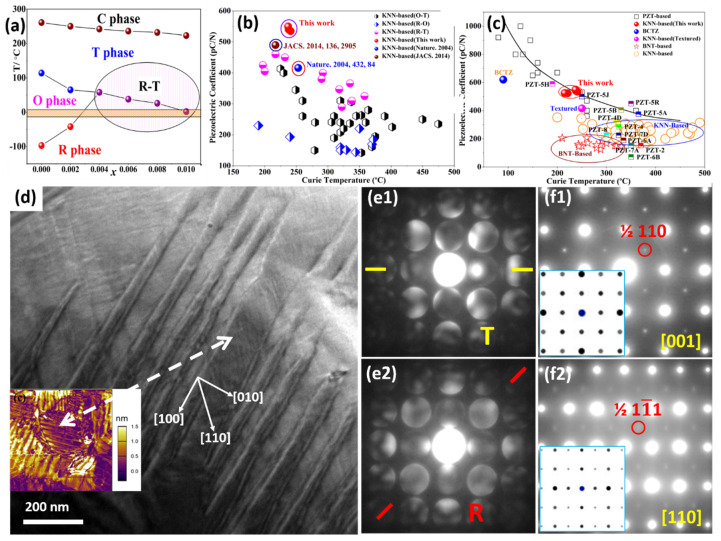
Piezoelectric property optimization via constructing the polymorphic *R*-*T* phase boundary in KNN-based materials: (**a**) phase diagram of KNN-based materials with respect to the fraction of BiFeO_3_. (**b**,**c**) comparison of piezoelectricity as a function of *T_C_* of the KNNS-xBF-yBNZ and other lead-based and lead-free piezoceramics. Hierarchical nanodomain structure of KNN-based piezoelectrics: (**d**) TEM image showing structures with hierarchical nanodomain, with a PFM photo inset. (**e1**,**e2**) CBED patterns of adjacent nanodomains, showing mirror planes along {010} and {100} for *R* phase and *T* phases, respectively. (**f1**,**f2**) Electron diffraction patterns showing superlattices, with simulated patterns inset. Reproduced from: Ref. [44] Copyright 2016 American Chemical Society; Ref. [114] Copyright 2017 Royal Society of Chemistry.

**Figure 2 materials-15-00487-f002:**
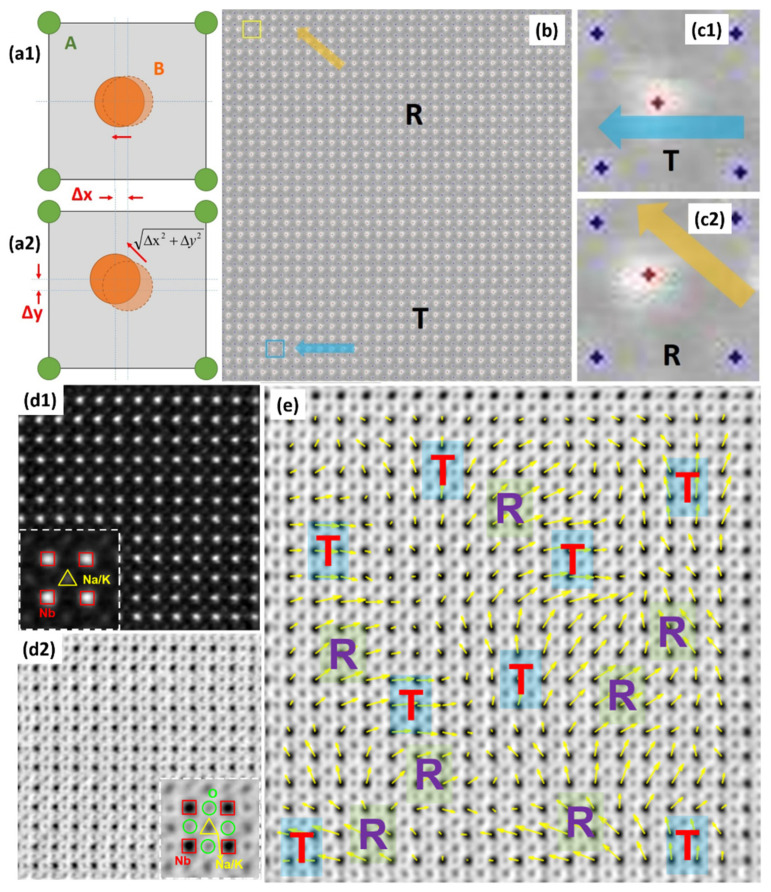
Local symmetry in nanodomains. (**a1**,**a2**) Schematic showing R and T symmetries with B atom displacement along the cube diagonal and pseudo cubic axes with respect to A (Na/K/Bi) atoms. (**b**) STEM HAADF lattice image on the domain boundary after peak finding, indicating the R and T phases. (**c1**,**c2**) Enlarged photos of the regions in (**b**) within the yellow box, showing R and T symmetries. Reproduced from: Ref. [44] Copyright 2016 American Chemical Society. Phase coexistence with gradual polarization rotation. (**d1**,**d2**) Simultaneously acquired STEM HAADF and ABF images at a domain boundary, with enlarged images inset. (**e**) Atomic displacement vector maps of centre Nb columns with respect to O columns.

**Figure 3 materials-15-00487-f003:**
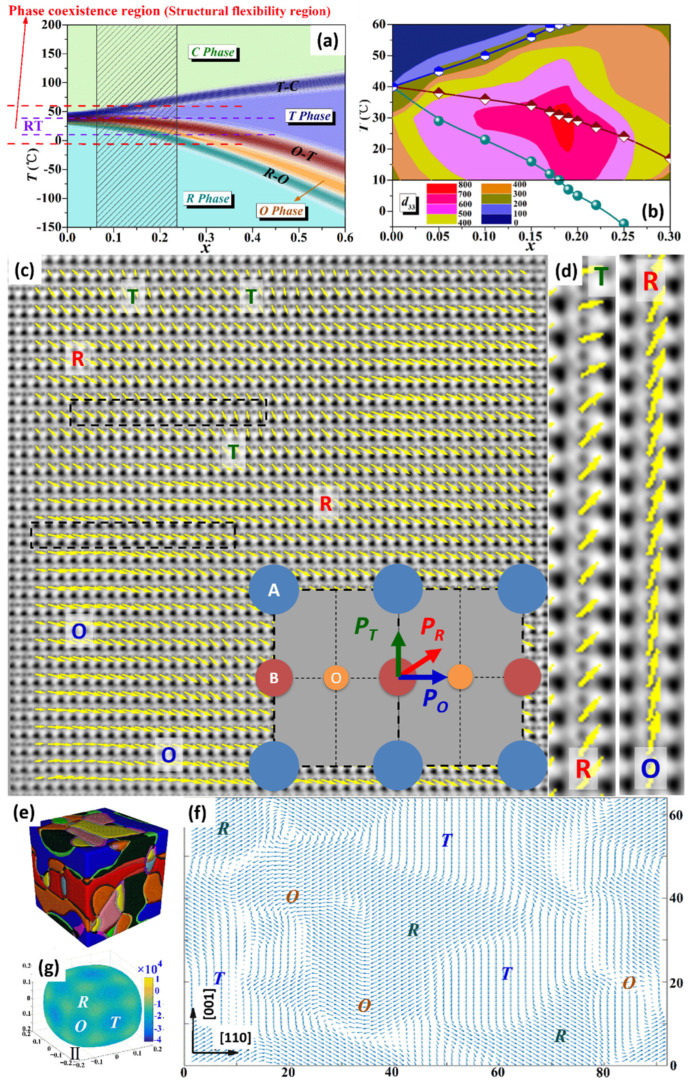
Practical high piezoelectricity in BaTiO_3_-based piezoelectric material using a quadruple critical point. (**a**) Schematic showing phase coexistence regimes based on phase diagram of (1−*x*)Ba(Ti_0.89_Sn_0.11_)O_3_−*x*(Ba_0.7_Ca_0.3_)TiO_3_ (BTS-BCT). Three dark stripes represent the *R*-*O*, *O*-*T*, and *T*-*C* polycrystalline phase transition regions, respectively. The region between red lines represents the phase coexistence region formed by the overlap of three bands. The shaded area shows the corresponding composition of the phase coexistence region of the high piezoelectric coefficient *d_33_* at indoor temperature. (**b**) Contour map of *d*_33_ coefficient in the temperature-composition plane for BTS-*x*BCT. Atomic-resolution polarization mapping for phase coexistence and gradual polarization rotation. (**c**) The *δ*_Ti-O_ displacement vector maps based on atomically-resolved STEM ABF image, the displacement vectors are indicated as arrows, the *T*, *O* and *R* regions are marked, and the inset is schematic projection of ABO_3_ unit cell along the {110} region axis, polarization directions for the *T*, *O* and *R* phases are marked accordingly. (**d**) Enlarged image showing continuous polarization rotation from *R* to *T* and from *O* to *R*. Theoretical calculation of BaTiO_3_ with multiphase coexistence and simulation for phase coexistence and continuous polarization rotation. (**e**) Phase-field simulations of three phase coexistence. (**f**) The projection for polarizations of (**e**) on the {110} plane. (**g**) Free-energy profiles of QCP. Reproduced from ref [127].

**Figure 4 materials-15-00487-f004:**
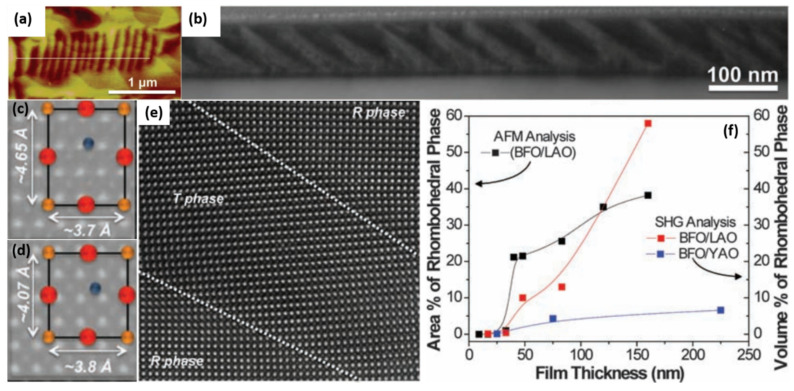
Nanoscale phase coexistence in BFO thin film at strain-driven phase boundary. (**a**) AFM image of nanodomains. (**b**) Low-magnification cross-section TEM image of nanodomains of BFO film. (**c**,**d**) Enlarged STEM HAADF images with schematic illustration of the unit cells for *T* phase and the *R* phase. (**e**) Atomically-resolved STEM HAADF of the boundaries between R and T regions. (**f**) The volume fraction of the *R* phase with thickness. Reproduced from Ref. [141]. Copyright 2009 American Association for the Advancement of Science.

**Figure 5 materials-15-00487-f005:**
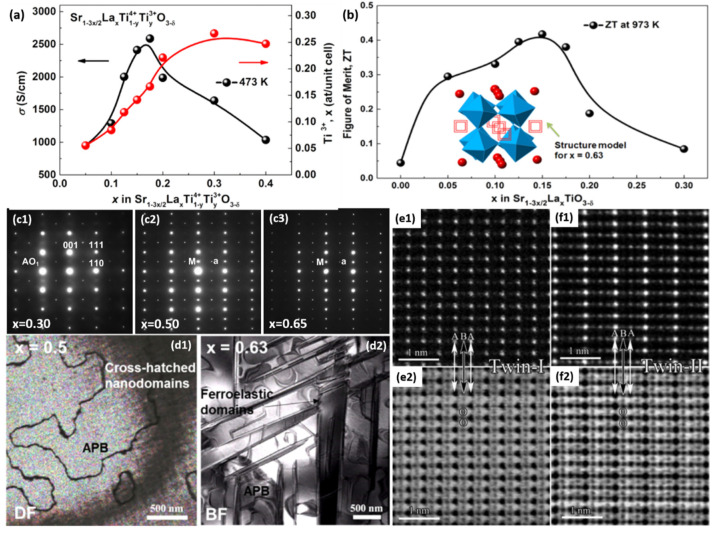
Vacancy ordering in SrTiO3-based thermoelectric oxide and influence on thermoelectric properties. (**a**) σ versus *x* at 473 K (black line) and Ti^3+^ content versus *x* in Sr_1−3*x*/2_La*_x_*TiO_3-δ_ ceramics sintered at 1773 K (red line) for 6 h in N_2_/5% H_2_. (**b**) *ZT* at 973 K versus *x* in Sr_1−3*x*/2_La*_x_*TiO_3-δ_ ceramics. (**c1**–**c3**) {110} zone axis diffraction patterns from Sr_1−3*x*/2_La*_x_*TiO_3_ ceramics; Superstructure reflections are indicated. (**d1**,**d2**) Dark-field and bright-field TEM images of Sr_1−3*x*/2_La*_x_*TiO_3_ for ceramics with *x* = 0.50 and 0.63. (**e1,e2**), and (**f1,f2**) atomic resolution STEM HAADF/ABF images of two domains of Sr_1−3*x*/2_La*_x_*TiO_3_ (*x* = 0.63) ceramic, showing A-site vacancy ordering along the {100} direction. Reproduced from Ref. [146] Copyright 2016 American Chemical Society.

**Figure 6 materials-15-00487-f006:**
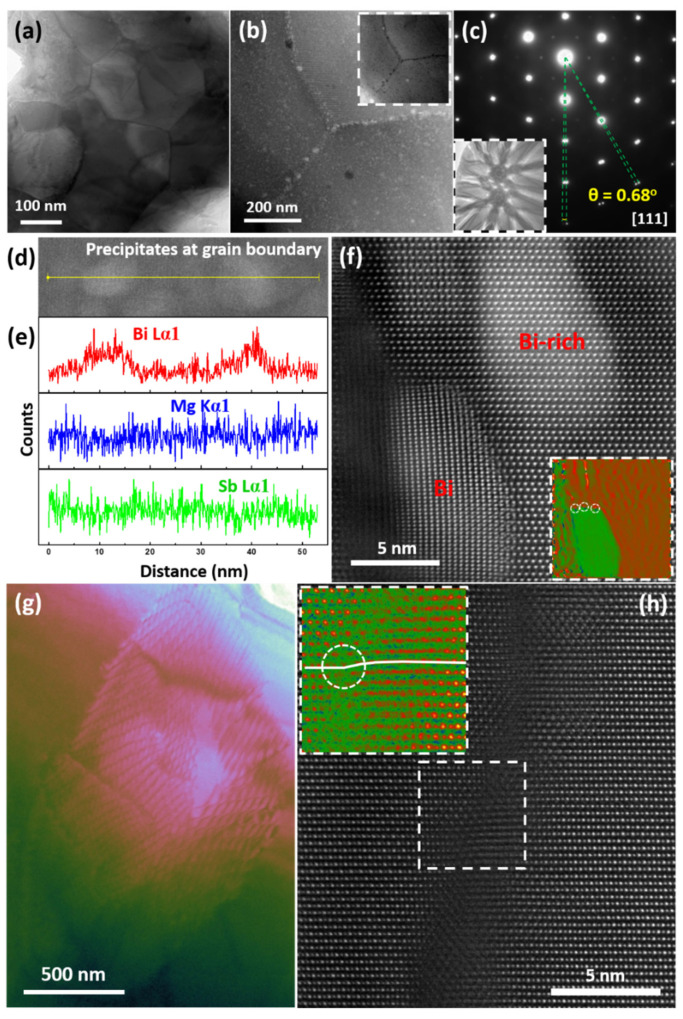
Grain boundaries and stacking faults of thermoelectric materials. (**a**) Low magnification TEM image showing submicron grains. (**b**) STEM HAADF image of a grain boundary triple-junction with segregated precipitates, with simultaneously acquired STEM ABF image inset. (**c**) Electron diffraction pattern from the grain boundary of SnTe-based sample along {111}, with STEM Rochigram pattern inset. (**d**,**e**) STEM HAADF image of segregated precipitates at a grain boundary and respective energy dispersive X-ray spectroscopy (EDS) line profiles of Bi, Mg and Sb for the Mg_3_Sb_1.5_Bi_0.5_ sample. Reproduced from Ref. [167] Copyright 2018 Elsevier. (**f**) STEM HAADF image of precipitates on the grain boundary, with its GPA analysis inset. (**g**) TEM images showing compact stacking faults of n-type PbTe based materials. (**h**) High-resolution STEM HAADF image from one stacking fault, with enlarged images of the marked region inset. Reproduced from Ref. [104] Copyright 2018 Royal Society of Chemistry.

**Figure 7 materials-15-00487-f007:**
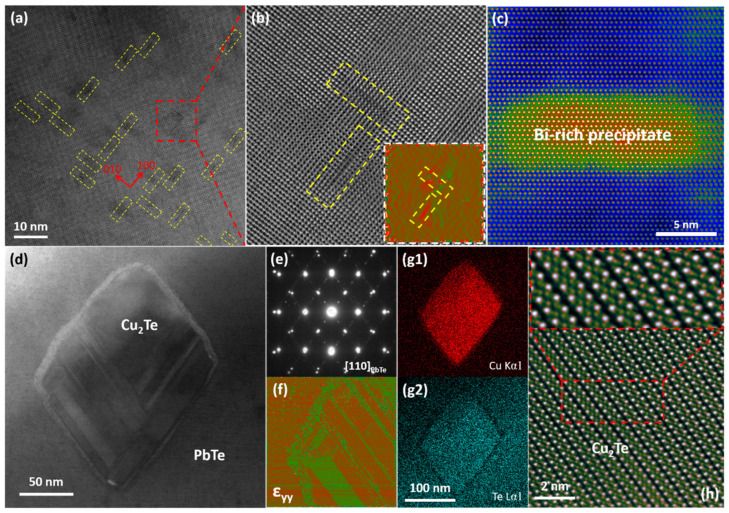
Nanoscale precipitates of thermoelectric materials. (**a**) HRTEM image of platelet-like nanostructures. (**b**) Enlarged lattice image of two perpendicular platelet-like nanostructures, with GPA strain analysis inset. Reproduced from Ref. [4] Copyright 2017, American Chemical Society. (**c**) STEM HAADF image of one Bi-rich precipitate with Mg_3_Sb_2_ based matrix. Reproduced from Ref. [167] Copyright 2018, Elsevier. (**d**) Low-mag STEM ABF image of a cubic Cu_2_Te precipitate within PbTe; (**e**) Electron diffraction patterns along {110} obtained at the interface between Cu_2_Te precipitate and PbTe matrix. (**f**) Fast Fourier transformation (FFT) image and GPA strain analysis of a Cu_2_Te-PbTe interface. (**g1**,**g2**) EDS elemental mapping of the Cu_2_Te precipitate. (**h**) Obtained STEM HAADF image of Cu_2_Te layered structure simultaneously, with amplified images around the stacking faults inset. Reproduced from Ref. [4] Copyright 2017 American Chemical Society.

**Figure 8 materials-15-00487-f008:**
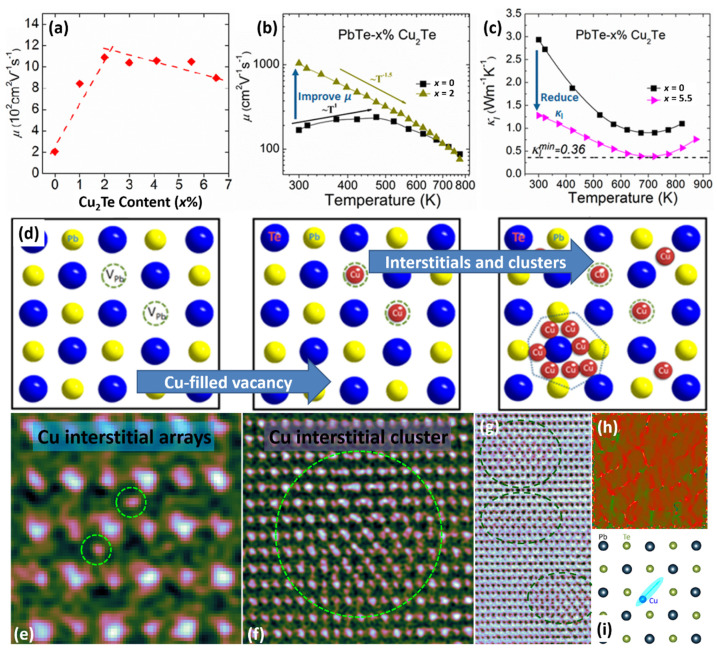
Synergistic role of copper in achieving high thermoelectric performance of n-type PbTe-Cu_2_Te. (**a**) Experimental and calculated carrier concentration and (**b**) carrier mobility as a function of Cu_2_Te. (**c**) Carrier mobility and lattice thermal conductivity are improved simultaneously through introducing Cu into the n-type PbTe. (**d**) Schematic of Cu atom occupying Pb vacancy. (**d**) The intrinsic Pb vacancies were firstly filled with Cu atoms to enhance carrier mobility, and then the excess Cu atoms formed voids and precipitations to decrease the thermal conductivity of the lattice. Atomic-scale defects for optimizing electrical and thermal transport via introducing Cu into n-type PbTe. (**e**,**f**) Atomically-resolved STEM HAADF images (colorized) showing Cu interstitial arrays and Cu interstitial clusters. (**g**) Atomically-resolved STEM ABF image showing three Cu interstitial clusters. (**h**) GPA strain analysis. (**i**) Schematic of a Cu atom vibrating around a Pb vacancy. Reproduced from Ref. [4] Copyright 2017 American Chemical Society.

**Figure 9 materials-15-00487-f009:**
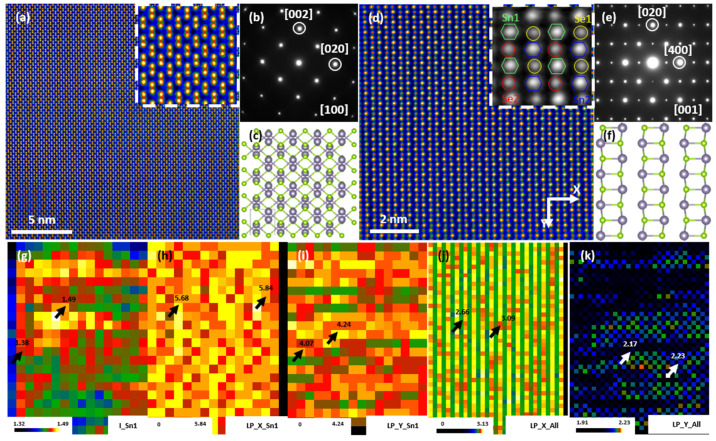
Substitutions in Te-alloyed SnSe crystals. (**a**) Atomically-resolved STEM HAADF image along {100} region axis, embedded with magnified images. (**b**,**c**) Electron diffraction pattern and structural model along {100} zone axis. (**d**) Atomically-resolved STEM HAADF image along {001} region axis, embedded with magnified images. The four types of atomic columns are labeled Sn1, Sn2, Se1 and Se2. (**e**,**f**) Electron diffraction pattern and structural model along {001} axis. (**g**–**j**) Intensity and lattice parameter distribution of different types of atom columns obtained from (**d**), where the lattice parameters unit is Å and intensity values are relative. (**g**) Intensity mapping of Sn1 atom columns. (**h**) Lattice parameters (LP) of Sn1 atom columns along X direction. (**i**) Lattice parameters of Sn1 atom columns along Y direction. (**j**,**k**) Lattice parameters of all the atom columns along X and Y directions. Reproduced from Ref. [33] Copyright 2019 American Chemical Society.

## Data Availability

Not applicable.

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
