# Peer review of "Seeing Structural Mechanisms of Optimized Piezoelectric and Thermoelectric Bulk Materials through Structural Defect Engineering"

_materials, 2022, doi:10.3390/ma15020487_

Round 1

Reviewer 1 Report

REFEREE’S REPORT: "Seeing structural origins and foreseeing new pathways to improved piezoelectric and thermoelectric bulk materials with aberration-corrected STEM ".

The present review is raises important task in material science field. Precise analysis and control of the defect chemistry, their control effect on the physical properties of selected classes of materials is a crucial milestone for the implementation of materials into application. The review presents the results of the authors, as well as studies of well-known scientific groups in relevant fields.

However, I have some remarks:

1. The main issue - the organization of figures and their discussion in the main text:

Figure 1. In the main text, there is no mention of Fig.1c.

Figure 10. Figure 10 (b1 and b2) is considered in the text, but these panels are missing in Figure 10.

Figure 11. In the text, Figure 11c, which does not correspond to the panel of Fig.11c.

Figure 11. The caption to Fig.11 refers to (h), which is not on the main panel of Fig.11, and Fig.11d does not show the mobility of carriers as a function of Cu2Te.

Figure 13. The text discusses Figure 13 (g1), which is not in Figure 13.

Figure 13. In the caption to Fig.13 refer (i1, i2, h1, h2, j1, j2), which are not on the main panel of Fig.13.

2. on page 19 (line 459): The authors refer to Fig. 6 (j, k), which do not correspond to the points discussed in the text.

3. Read the text carefully, there are sentences without logical ending, for example:

Atomic scale defects are always thermodynamic.

The key to increasing thermoelectric efficiency is the dynamics of atoms and their defects in lattices.

4. Please update the Reference [128].

This review provides valuable experimental information it could be recommended for the publication in Materials after minor revision.

Author Response

Reviewer #1

REVIEWER’S REPORT: "Seeing structural origins and foreseeing new pathways to improved piezoelectric and thermoelectric bulk materials with aberration-corrected STEM ".

General comment 1: The present review is raises important task in material science field. Precise analysis and control of the defect chemistry, their control effect on the physical properties of selected classes of materials is a crucial milestone for the implementation of materials into application. The review presents the results of the authors, as well as studies of well-known scientific groups in relevant fields.

General response: Thank the Reviewer for the evaluation and for his/her valuable suggestions.

Detailed comment 1: The main issue - the organization of figures and their discussion in the main text:

  • Figure 1. In the main text, there is no mention of Fig.1c.
  • Figure 10. Figure 10 (b1 and b2) is considered in the text, but these panels are missing in Figure 10.
  • Figure 11. In the text, Figure 11c, which does not correspond to the panel of Fig.11c.
  • Figure 11. The caption to Fig.11 refers to (h), which is not on the main panel of Fig.11, and Fig.11d does not show the mobility of carriers as a function of Cu2
  • Figure 13. The text discusses Figure 13 (g1), which is not in Figure 13.
  • Figure 13. In the caption to Fig.13 refer (i1, i2, h1, h2, j1, j2), which are not on the main panel of Fig.13.

Response: Thank the Reviewer for scrupulously checking our manuscript. We revised the figures and the related text carefully in the revised manuscript.

Detailed comment 2: on page 19 (line 459): The authors refer to Fig. 6 (j, k), which do not correspond to the points discussed in the text.

Response: Thanks. It should be Figures 13(h-j) in the previous version. We revised it in the revised manuscript.

Detailed comment 3: Read the text carefully, there are sentences without logical ending, for example:

Atomic scale defects are always thermodynamic.

The key to increasing thermoelectric efficiency is the dynamics of atoms and their defects in lattices.

Response: We revised them to:

“Atomic scale defects are always in thermodynamic states.”

“The key to increasing thermoelectric efficiency is to manipulate the dynamics of atoms and their defects in lattices.”

Detailed comment 4: Please update the Reference [128].

Response: We have updated it in the revised manuscript.

Zhao, C.; Wu, H.; Li, F.; Cai, Y.; Zhang, Y.; Song, D.; Wu, J.; Lyu, X.; Yin, J.; Xiao, D.; Zhu, J.; Pennycook, S. J., Practical High Piezoelectricity in Barium Titanate Ceramics Utilizing Multiphase Convergence with Broad Structural Flexibility. Journal of the American Chemical Society 2018, 140, (45), 15252-15260.

General comment 2: This review provides valuable experimental information it could be recommended for the publication in Materials after minor revision.

General response: Thank the Reviewer for the evaluation and for his/her valuable suggestions.

Reviewer 2 Report

This paper reviews the application of aberration-corrected scanning transmission electron microscopy in characterizing piezoelectric and thermoelectric materials. The experimental scope is sharply focused on microscopy, while the materials for which this method is applicable are discussed in the paper. As a result, this paper can be an essential reference for materials scientists wanting a quick and concise guide to characterization. While the paper contains many primary citations, the reference list can be further enhanced by including more recent literature from aligned topics to direct interested readers. The following are a few suggestions:

When discussing equations 1, 2, and 3, it would be beneficial to readers to refer to the following recent reference, which elaborates on these equations computationally. After all, in recent times, these equations are certainly solved computationally for any given material, especially when the investigated material is doped or defective:

Gorai et al. Computationally guided discovery of thermoelectric materials, Nature Review Materials 2, 17053, 2017. https://doi.org/10.1038/natrevmats.2017.53

On page 15, when discussing the layered thermoelectric systems, the following recent and comprehensive paper can be insightful to the readers:

Y Liu et al. Recent advances of layered thermoelectric materials, Advanced Sustainable Systems 2, 1800046, 2018, https://doi.org/10.1002/adsu.201800046

For the discussions of Cu dopant reducing the thermal PbTe conductivity (page 17), the following reference provides an in-depth experimental and theoretical review and discussions, which may be interesting to curious readers:

H.A. Eivari et al. Low thermal conductivity: fundamentals and theoretical aspects in thermoelectric applications, Materials Today Energy 21, 100744, 2021, https://doi.org/10.1016/j.mtener.2021.100744

Lastly, one minor point: The in-line mathematical notation on page 10 needs a desperate brush-up. For instance, in some of the notations, the h bar is poorly produced. The sub- and superscripts seem to be out of alignment.

Author Response

Reviewer #2

­­General comment:  This paper reviews the application of aberration-corrected scanning transmission electron microscopy in characterizing piezoelectric and thermoelectric materials. The experimental scope is sharply focused on microscopy, while the materials for which this method is applicable are discussed in the paper. As a result, this paper can be an essential reference for materials scientists wanting a quick and concise guide to characterization.

General response: Thank the Reviewer for the evaluation and for his/her valuable suggestions.

Detailed comment 1: While the paper contains many primary citations, the reference list can be further enhanced by including more recent literature from aligned topics to direct interested readers.

Response: We have added several recent citations in the revised manuscript.

Detailed comment 2:

The following are a few suggestions:

When discussing equations 1, 2, and 3, it would be beneficial to readers to refer to the following recent reference, which elaborates on these equations computationally. After all, in recent times, these equations are certainly solved computationally for any given material, especially when the investigated material is doped or defective:

Gorai et al. Computationally guided discovery of thermoelectric materials, Nature Review Materials 2, 17053, 2017. https://doi.org/10.1038/natrevmats.2017.53

On page 15, when discussing the layered thermoelectric systems, the following recent and comprehensive paper can be insightful to the readers:

Y Liu et al. Recent advances of layered thermoelectric materials, Advanced Sustainable Systems 2, 1800046, 2018, https://doi.org/10.1002/adsu.201800046

For the discussions of Cu dopant reducing the thermal PbTe conductivity (page 17), the following reference provides an in-depth experimental and theoretical review and discussions, which may be interesting to curious readers:

H.A. Eivari et al. Low thermal conductivity: fundamentals and theoretical aspects in thermoelectric applications, Materials Today Energy 21, 100744, 2021, https://doi.org/10.1016/j.mtener.2021.100744

Response: Thank the Reviewer #2 for his kind suggestions on related literatures. We have added these references in the revised manuscript.

Gorai et al contributed a great review on the computationally guided discovery of ther-moelectric materials.[159]

These layered structures could effectively scatter phonons across various length scales and lead to extremely low lattice thermal conductivity, Figure 8(c).[4, 170]

Cu atoms cause the local lattice disorder, which play an important role in the scattering of phonons with high frequency at high temperatures.[4, 43, 171]

Detailed comment 3: Lastly, one minor point: The in-line mathematical notation on page 10 needs a desperate brush-up. For instance, in some of the notations, the h bar is poorly produced. The sub- and superscripts seem to be out of alignment.

Response: We have revised the notations on page 10 in the revised manuscript.

Reviewer 3 Report

Current manuscript is submitted as a review with a title promising to foresee new pathways to improved piezoelectric and thermoelectric bulk materials with aberration-corrected STEM. STEM is indeed shown to be a powerful technique for observation of structural features and defects. However, no new pathways are given in the manuscript by its authors despite of many words “new” in the abstract. Enhancement of the piezoelectric coefficient by morphotropic phase boundary is known for decades. Suppression of thermal conductivity in thermoelectrics by nanostructuring and defect formation are also evident for a while (see e.g. review in C 2021, 7, 37), whereas their effect on the electrical conductivity and Seebeck coefficient is not that unambiguous. However, current work do not discuss Seebeck coefficient at all, concluding just that “atomic-scale point defect engineering will become a new strategy to improve both electrical and thermal properties of thermoelectric materials”, despite that it is reported and presented at least 5 years ago in e.g. Chem. Mater. 2016, 28, 925 but not mentioned in this manuscript.

Another drawback of the manuscript is a lack of connection between the perovskite oxide piezoelectrics and mainly chalcogenide-type thermoelectrics discussed in this manuscript. Oxide thermoelectrics could thus be a perfect link, which however is currently missing. Moreover, the manuscript looks to be based on only four publications (Refs. 4, 33, 44, 128) with contribution of the corresponding author. As a result, the manuscript is not only unconnected but also one-sided, being written not as an independent review but as an article without experimental part but with many statements not appropriate for review such as we did this, we did that. It looks unlikely that nobody else have applied STEM to study piezoelectrics and thermoelectrics, and thus a significant manuscript revision is required. Furthermore, one of the core references, Ref. 128, indicating just “Submitted, 2018” leads the reader to nothing. It is thus impossible to find the experimental, simulation and other details behind Figures 5-7 of the current manuscript. That is not acceptable.

As to the references, they have to be a base for review, but in current manuscript they are not formatted to the Materials style, they do not show article titles, they are incomplete or incorrect in the case of Refs. 7, 8, 14, 19, 21, 28, 49, 53, 69, 70, 74, 86, 92, 101, 104, 105, 106, 125, 144, besides absolutely useless Ref. 128. In addition, Refs. 67 and 68 look to cite the same work.

The language of the review manuscript does not allow the reader to follow it easily as well. First of all, it does not always sound scientifically. It has many words “improve(e/ed/ing/ment)”, which are scientifically ambiguous depending on the point of view. Just for instance, the authors wrote in the caption of Figure 1 Piezoelectric property improvement via optimizing the polymorphic R-T phase boundary in KNN-based materials.” However, Figure 1, showing enhanced piezoelectric coefficient, presents as well decreased Curie temperature that is not an improvement. Then the authors incorrectly call piezoelectric solid solutions as alloys, although piezoelectric oxides are not metals and were not melted to form the solid solution. Twice the authors incorrectly call relaxation time as relax time and once mass as mess. It is also unclear what external field is meant at line 34, what means (j) at line 235, EDX at line 304, FFT at lines 353 and 409. Figure 1b shows no dashed lines in contrast to the caption. There is also a big confusion with Figure 11, its caption and description in the text, which do not correspond to each other, but still tend to “improve” something. In less degree but the same lack of correspondence is valid for Figure 13.

English of the manuscript is also often incorrect starting from the very first sentence of the Introduction section. As a result, many verbs are used incorrectly across the text, and sentences at lines 67-71, 114-120, 343-345, 450-451 and 455-456 have to be revised to clarify the message.

Thus, current review manuscript does not correspond to its title, avoids results published by anybody except the corresponding author, presents two unconnected parts with useless reference as a base for one of them and many incorrect/incomplete references in general, besides numerous scientific and English grammar mistakes. That makes it incapable to be recommended for publication.

Author Response

Reviewer #3

­­General comment 1:  Current manuscript is submitted as a review with a title promising to foresee new pathways to improved piezoelectric and thermoelectric bulk materials with aberration-corrected STEM. STEM is indeed shown to be a powerful technique for observation of structural features and defects.

Response: Thank the Reviewer for the evaluation and for his/her valuable suggestions.

Detailed comment 1: However, no new pathways are given in the manuscript by its authors despite of many words “new” in the abstract. Enhancement of the piezoelectric coefficient by morphotropic phase boundary is known for decades.

Response: Seeing is believing. Interstitial point defect is one example demonstrating how to foresee new pathways to improved functional materials through aberration-corrected STEM. Nanostructuring has been widely acknowledged as the most universal strategy to enhance the thermoelectrc properties. However, the atomic-scale defects are always ignored due to its ultra-small size, invisible via traditional methods, thus most researches have stayed at the assumption stage. With the new generation of aberration-corrected microscopes this has now become possible, which is the main toipic of the present review.  The recent achievements have been able to directly observe the intrinsic vacancies and extrinsic interstitials and substitutions in most thermoelectric systems. Especially for interstitials, such type of point defects has become the brand-new strategy for synergistic optimization of phonon and carrier transport. During this progress, direct seeing the point defects and revealing the structure-property correlation play important roles.

It is true that enhancement of the piezoelectric coefficient by morphotropic phase boundary has been known for decades, however, the origin is still under controversial. New types of phase boundaries have been proposed and employed to further enhance the piezoelectric properties, especially for lead-free systems. For example, the new type phase boundary with quadruple point for BaTiO3-based lead-free piezoelectric system was introduced, which was actually based on the “seeing” the structural origin through aberration-corrected STEM, please see Figure 5-7 in the previous version, or Figure 3 in the revised version.

Detailed comment 2:  Suppression of thermal conductivity in thermoelectrics by nanostructuring and defect formation are also evident for a while (see e.g. review in C 2021, 7, 37), whereas their effect on the electrical conductivity and Seebeck coefficient is not that unambiguous. However, current work do not discuss Seebeck coefficient at all, concluding just that “atomic-scale point defect engineering will become a new strategy to improve both electrical and thermal properties of thermoelectric materials”, despite that it is reported and presented at least 5 years ago in e.g. Chem. Mater. 2016, 28, 925 but not mentioned in this manuscript.

Response: Thanks for suggesting the new citation (C 2021, 7, 37) about nanostructures. We cited and discussed it in the main text, as well the other related paper (Okhay, O et al., Carbon 2019, 143, 215–222).

For example, Okhay contributed a good review about impact of graphene or reduced graphene oxide on performance of thermoelectric composites, including chalcogenides, skutterudites, and metal oxides.[159, 160]

Thanks for introducing the great paper (Chem. Mater. 2016, 28, 925), which is highly related with the present review. We have highlighted this work with a new whole figure (See Figure 5) in the revised manuscript, as a representative example to show how structural defects contribute to both thermal and electrical transport properties.

Figure 5. Vacancy ordering in SrTiO3-based thermoelectric oxide and influence on thermoelectric properties. (a) σ versus x at 473 K and Ti3+ content versus x in Sr1–3x/2LaxTiO3-δ ceramics sintered at 1773 K for 6 h in N2/5% H2. (b) ZT at 973 K versus x in Sr1–3x/2LaxTiO3-δ ceramics. (c1-c3) ⟨110⟩ zone axis diffraction patterns from Sr1–3x/2LaxTiO3 ceramics; Superstructure reflections are indicated. (d1, d2) Dark-field and bright-field TEM images of Sr1–3x/2LaxTiO3 for ceramics with x = 0.50 and 0.63. (e1, e2) and (f1, f2) Atomic resolution STEM HAADF/ABF images of two domains of Sr1–3x/2LaxTiO3 (x = 0.63) ceramic, showing A-site vacancy ordering along the ⟨100⟩ direction. Reproduced from ref.[143] Copyright 2016, American Chemical Society.

Detailed comment 3: Another drawback of the manuscript is a lack of connection between the perovskite oxide piezoelectrics and mainly chalcogenide-type thermoelectrics discussed in this manuscript. Oxide thermoelectrics could thus be a perfect link, which however is currently missing.

Response: Thanks for the Reviewer’s valuable suggestion. We have added one part (See Section 3. Ferroelectric-Thermoelectrics: bridge between piezoelectrics and thermoeelectrics) about oxide thermoelectrics in the revised manuscript, so as to connect the perovskite oxide piezoelectrics and mainly chalcogenide-type thermoelectrics.

Detailed comment 4: Moreover, the manuscript looks to be based on only four publications (Refs. 4, 33, 44, 128) with contribution of the corresponding author. As a result, the manuscript is not only unconnected but also one-sided, being written not as an independent review but as an article without experimental part but with many statements not appropriate for review such as we did this, we did that. It looks unlikely that nobody else have applied STEM to study piezoelectrics and thermoelectrics, and thus a significant manuscript revision is required.

Response: We have added two figures from others in the revised manuscript, and one is suggested by the Reviewer, see Figure 5 in the revised version, the other is from other researchers, see Figure 4 in the revised version. Moreover, we have simplified the previous 13 figures to 7 figures, please see Figures 1-3,7,8 in the revised version.

We revised the in appropriate statements in the revised manuscript.

Detailed comment 5: Furthermore, one of the core references, Ref. 128, indicating just “Submitted, 2018” leads the reader to nothing. It is thus impossible to find the experimental, simulation and other details behind Figures 5-7 of the current manuscript. That is not acceptable.

Response: We have updated it in the revised manuscript.

Detailed comment 6: As to the references, they have to be a base for review, but in current manuscript they are not formatted to the Materials style, they do not show article titles, they are incomplete or incorrect in the case of Refs. 7, 8, 14, 19, 21, 28, 49, 53, 69, 70, 74, 86, 92, 101, 104, 105, 106, 125, 144, besides absolutely useless Ref. 128. In addition, Refs. 67 and 68 look to cite the same work.

Response: In the revised manuscript, we have formatted references to Materials style and maken these references complete. We removed the former Ref. 67.

Detailed comment 7: The language of the review manuscript does not allow the reader to follow it easily as well. First of all, it does not always sound scientifically. It has many words “improve(e/ed/ing/ment)”, which are scientifically ambiguous depending on the point of view. Just for instance, the authors wrote in the caption of Figure 1 Piezoelectric property improvement via optimizing the polymorphic R-T phase boundary in KNN-based materials.” However, Figure 1, showing enhanced piezoelectric coefficient, presents as well decreased Curie temperature that is not an improvement. Then the authors incorrectly call piezoelectric solid solutions as alloys, although piezoelectric oxides are not metals and were not melted to form the solid solution. Twice the authors incorrectly call relaxation time as relax time and once mass as mess. It is also unclear what external field is meant at line 34, what means (j) at line 235, EDX at line 304, FFT at lines 353 and 409. Figure 1b shows no dashed lines in contrast to the caption. There is also a big confusion with Figure 11, its caption and description in the text, which do not correspond to each other, but still tend to “improve” something. In less degree but the same lack of correspondence is valid for Figure 13.

Response: Thanks for the Reviewer Thank the Reviewer for scrupulously checking our manuscript. We revised these words in the revised manuscript.

 “improvement” mentioned here to “optimization”; “alloying” to “composition tuning”; “relax time” to “relaxation time”; “mess” to “mass”; “external field” to “external thermal/stress/electric field”; “(j)” at line 235 to “d”; “EDX” at line 304 to “EDS”; “FFT” at lines 353 to “fast Fourier transformation (FFT)”.

We deleted the description of “dashed lines” in the caption of the previous Figure 1b.

We have changed the captions and the discussions in the main text of the previous Figures 11 and 13.

Detailed comment 8: English of the manuscript is also often incorrect starting from the very first sentence of the Introduction section. As a result, many verbs are used incorrectly across the text, and sentences at lines 67-71, 114-120, 343-345, 450-451 and 455-456 have to be revised to clarify the message.

Response: We revised these words and sentences in the revised manuscript.

General comment 2:  Thus, current review manuscript does not correspond to its title, avoids results published by anybody except the corresponding author, presents two unconnected parts with useless reference as a base for one of them and many incorrect/incomplete references in general, besides numerous scientific and English grammar mistakes. That makes it incapable to be recommended for publication.

Response: Thank the Reviewer again for prudentially reviewing our manuscript and for his/her valuable suggestions. We have thoroughly revised the manuscript according to the Reviewer’s suggestions.

Round 2

Reviewer 3 Report

The authors have addressed the previous report comments in quite proper way, although not always. Therefore, the manuscript can be considered for publication, but there are still some revisions to be done.

First, words “foreseeing new” are still needless in the title, since they do not correspond indeed well to the manuscript body with extraordinary seeing but very limited foreseeing. Moreover, 3 words “new” at a single line 16 of the abstract look to be needless as well.

Second, adding overviews on results by other groups and providing a link between perovskite piezoelectric oxides and thermoelectric chalcogenides, the authors have introduced a section “3. Ferroelectric-thermoelectrics: the bridge between piezoelectrics and thermoelectrics”. However, that is not correctly done. La-doped SrTiO3 are perovskite oxides like piezoelectrics, but they are not ferroelectrics at all in contrast to the section title. Therefore, these results are suggested to be just first presented in the Thermoelectrics section rather than in a separate section with incorrect title. Moreover, the sentence at lines 267-269, if correct, has to be supported by references. Furthermore, Seebeck coefficient is not electrical transport characteristic as stated at line 304 and it is still not discussed anywhere else in the manuscript that once again present the manuscript limitation. In addition, stating at lines 392-393 that “phase boundaries can effectively scatter phonons and facilitate carrier transport.” the authors have to explain the facilitation mechanism.

Concerning the language, removing words “we”, words “our” have to be removed as well. Symbols and abbreviations like d33, d33*, Tc, FFT still have to be decoded. Moreover, EDS is first mentioned at line 359, but decoded just at line 389. Corrections of “(c1, d2) Enlarged photos of the regions in (h)” at line 160, “the phase boundary exists dislocation row” at line 395, “Our most recent results was” at line 414, “Figure 9(b)” at lines 479-480 and “To analysis Te substitution and bonds, the quantitative calculation” at line 485 still have to be done. Furthermore, writing at line 24 “For most crystalline materials”, the authors probably meant “for the majority of crystalline materials”, but what they mean by “materials with perfect crystals” at line 26 is still unclear. At lines 67-69, one should either remove word “While” or use comma instead of dot at the end. “responses … is” at lines 71-72 are also grammatically incorrect as well as “to tuning” at line 76 and “grains comes” at line 338. “optimization via optimizing” at line 106 does not also look well. Verb “remove” at line 301 should be rather “harvest”, while at line 326 “et al.” should be rather “etc.” At line 347 “precipitates segregated with triple junctions” should be “precipitates segregated at triple junctions. At line 378, “Okhay” should be corrected to “Okhay et al.” At line 322, kB should be with subscripted B, while at line 294 Cu2Te should be with subscripted 2.

Thus, the revised manuscript can be considered for publication, but only when the additional revisions mentioned above are done and the manuscript text double checked.

Author Response

General comment 1: The authors have addressed the previous report comments in quite proper way, although not always. Therefore, the manuscript can be considered for publication, but there are still some revisions to be done.

General response: Thank the Reviewer for carefully evaluation again and for his/her valuable suggestions.

Detailed comment 1: First, words “foreseeing new” are still needless in the title, since they do not correspond indeed well to the manuscript body with extraordinary seeing but very limited foreseeing. Moreover, 3 words “new” at a single line 16 of the abstract look to be needless as well.

Response: In the revised version, we changed “foreseeing new” to “foreseeing”; we deleted the 3 words “new” at line 16. 

Detailed comment 2: Second, adding overviews on results by other groups and providing a link between perovskite piezoelectric oxides and thermoelectric chalcogenides, the authors have introduced a section “3. Ferroelectric-thermoelectrics: the bridge between piezoelectrics and thermoelectrics”. However, that is not correctly done. La-doped SrTiO3 are perovskite oxides like piezoelectrics, but they are not ferroelectrics at all in contrast to the section title. Therefore, these results are suggested to be just first presented in the Thermoelectrics section rather than in a separate section with incorrect title. Moreover, the sentence at lines 267-269, if correct, has to be supported by references.

Response: In the revised version, we changed “foreseeing new” to “foreseeing”; we deleted the 3 words “new” at line 16. In some particular cases, SrTiO3 can be ferroelectrics, see Nature 430, 758–761 (2004). https://doi.org/10.1038/nature02773. and Nat Commun 11, 3141 (2020). https://doi.org/10.1038/s41467-020-16912-3. It is true that SrTiO3 ceramics are not ferroelectrics. We changed the title of this section to “Perovskite thermoelectric oxides: the bridge between piezoelectrics/ferroelectrics and thermoelectrics”, since we highly appreciate the Reviewer’s valuable suggestion in the 1st cycle of reviewing about the connection part between piezoelectrics and thermoelectrics. We also did some change in the main text, see below:

“However, weak piezoelectric/ferroelectric or even paraelectric oxides in the unusual condition where the concentration of electronic carriers is close to a metal–insulator transition have properties of interest for oxide based thermoelectric applications. There typical example is doped SrTiO3, which is a paraelectrics in ceramics whild could be ferroelectrics in certain strained condition.[60, 143, 144] In the heavily reduced, nonstoichiometric n-type perovskite SrTiO3−δ, it is shown that metallic-like conductivity happens in its paraelectric phase and the potential ferroelectricity could stabilize semiconducting feature.[145-148]”

We added the references for Line 267-269 in the revised version.

Detailed comment 3: Furthermore, Seebeck coefficient is not electrical transport characteristic as stated at line 304 and it is still not discussed anywhere else in the manuscript that once again present the manuscript limitation.

Response: Seebeck coefficient is one main electrical transport characteristic, as electrical conductivity. In thermoelectrics, the well-employed band structure engineering strategies to boost Seebeck coefficient, e.g., band alignment and band gap enlargement, are highly related with substitutions, doping with the same valence, like Sr doped in PbTe,[Nature Chem 3, 160–166 (2011). and Nature 489, 414–418 (2012)], Mg doped in PbTe,[Energy Environ. Sci., 2013,6, 3346-3355] and Se doped in PbTe.[Nature 473, 66–69 (2011)]. All these substitutions are point defects, one main structural defect discussed in the present review. To highlight it, we added these in the main text, see Page #15 of the revised version, or see below:

“In thermoelectrics, the well-employed band structure engineering strategies to boost Seebeck coefficient, e.g., band alignment and band gap enlargement, are highly related with substitutions. Here substitutions are doped with the same valence, like Sr doped in PbTe,[150, 151], Mg doped in PbTe,[13] and Se doped in PbTe.[10]

In the present review, we have present one example to show the importance of substitutions on band structure and thus on Seebeck coefficient, please see Figure 9. The related discussion is shown below:

“For SnSe, Te alloying as substitutions could increase the crystal symmetry, optimizes the bond structure, and changes the band shape, and thus enhance the electrical transport properties; meanwhile, the substitutions could play as phonon scattering cen-ters and contribute to lower the thermal conductivity. Thus, structural defects can also largely influence Seebeck coefficient.”

Detailed comment 4: In addition, stating at lines 392-393 that “phase boundaries can effectively scatter phonons and facilitate carrier transport.” the authors have to explain the facilitation mechanism.

Response: We changed this sentence to “These phase boundaries can effectively scatter phonons, without influencing carrier transport duo to little lattice mismatch between two phases” In the revised version.

Detailed comment 4: Concerning the language, removing words “we”, words “our” have to be removed as well. Symbols and abbreviations like d33, d33*, Tc, FFT still have to be decoded. Moreover, EDS is first mentioned at line 359, but decoded just at line 389. Corrections of “(c1, d2) Enlarged photos of the regions in (h)” at line 160, “the phase boundary exists dislocation row” at line 395, “Our most recent results was” at line 414, “Figure 9(b)” at lines 479-480 and “To analysis Te substitution and bonds, the quantitative calculation” at line 485 still have to be done. Furthermore, writing at line 24 “For most crystalline materials”, the authors probably meant “for the majority of crystalline materials”, but what they mean by “materials with perfect crystals” at line 26 is still unclear. At lines 67-69, one should either remove word “While” or use comma instead of dot at the end. “responses … is” at lines 71-72 are also grammatically incorrect as well as “to tuning” at line 76 and “grains comes” at line 338. “optimization via optimizing” at line 106 does not also look well. Verb “remove” at line 301 should be rather “harvest”, while at line 326 “et al.” should be rather “etc.” At line 347 “precipitates segregated with triple junctions” should be “precipitates segregated at triple junctions. At line 378, “Okhay” should be corrected to “Okhay et al.” At line 322, kB should be with subscripted B, while at line 294 Cu2Te should be with subscripted 2.

Response: Thanks again for carefully checking our manuscript. We have revised all according to the Reviewer’s suggestions in the revised version, please see the red marked sentences.

“materials with perfect crystals” has been changed to “materials with ideally perfect crystals”. It means such material crystal do not has any structural defect, but it is an ideal case. 

General comment 2: Thus, the revised manuscript can be considered for publication, but only when the additional revisions mentioned above are done and the manuscript text double checked.

General response: Thanks again. We have revised the manuscript carefully.

Round 3

Reviewer 3 Report

The previous report comments were addressed by the authors worse than the comments of the report before, resulting in still wrong scientific statements and significant mistakes.

Parameters d33 and d33* are usually called piezoelectric coefficient and effective piezoelectric coefficient but not piezoelectric constant and converse piezoelectric coefficient as did by the authors. As for the review paper, the authors have to be more rigorous in terminology.

Moreover, addressing the previous report comment, the authors wrote at lines 269-271 that “metallic-like conductivity happens in its paraelectric phase and the potential ferroelectricity could stabilize semiconducting feature.[145-148]” However, none of the Refs. 145-148 mentions neither paraelectric phase nor ferroelectricity in contrast to the authors’ statement. Moreover, this statement contradicts not only to the literature but even to the sentence written by the authors 10 lines above as ”When ferroelectrics become highly conductive, it destabilizes the long range dipolar ordering necessary for ferroelectricity.” Thus, the statement at lines 269-271 is apparently wrong and has to be revised accordingly, avoiding mixing ferroelectric and thermoelectric features, since they indeed do not enhance each other, despite being observed in materials with the same structure or even composition.

Furthermore, the previous sentence “There typical example is doped SrTiO3, which is a paraelectrics in ceramics whild could be ferroelectrics in certain strained condition.[60, 143, 144]” is full of grammatical mistakes, while the authors were asked to double check the manuscript for them. In addition, if the authors want to mention ferroelectricity induced in SrTiO3, they should also refer the recent review paper in Condens. Matter 2020, 5, 58.

The meaning of sentence at line 474 “Here substitutions are doped with the same valence” is also very unclear, requiring significant revision in order to pass the message.

Then, “To analysis” at line 490 should be rather “To assess”, “duo” at line 395 should be “due”, “(c1, d2)” at line 160 “(c1, c2)”, while “materials with ideally perfect crystals” at line 25 just “perfect crystalline materials”.

Last but not least, there is still no reason to keep the word “foreseeing” in the manuscript title, since approaches are observed, confirmed, approved, but not foreseen in this manuscript.

Author Response

General comment 1: The previous report comments were addressed by the authors worse than the comments of the report before, resulting in still wrong scientific statements and significant mistakes.

General response: Thank the Reviewer for carefully evaluation again and for his/her valuable suggestions. It is open to discuss the scientific understanding.

Detailed comment 1: Parameters d33 and d33* are usually called piezoelectric coefficient and effective piezoelectric coefficient but not piezoelectric constant and converse piezoelectric coefficient as did by the authors. As for the review paper, the authors have to be more rigorous in terminology.

Response: In the revised version, we changed “foreseeing new” to “foreseeing”; we deleted the 3 words “new” at line 16.

Sometimes, d33 and d33* are called piezoelectric coefficient and effective piezoelectric coefficient, while sometimes, they are called piezoelectric constant and converse piezoelectric constant, please see ACS Appl. Mater. Interfaces 2021, 13, 7461−7469. There is no difference between “coefficient” and “constant”. We prefer to “converse”, since it could reflect how the measurement more directly than “effective”, since d33* is obtained through the strain normalized by the given electric fields, which is converse piezoelectric effect.

Anyway, we changed to “effective piezoelectric coefficient” in the revised version.

Detailed comment 2: Moreover, addressing the previous report comment, the authors wrote at lines 269-271 that “metallic-like conductivity happens in its paraelectric phase and the potential ferroelectricity could stabilize semiconducting feature.[145-148]” However, none of the Refs. 145-148 mentions neither phase nor ferroelectricity in contrast to the authors’ statement.

Moreover, this statement contradicts not only to the literature but even to the sentence written by the authors 10 lines above as ”When ferroelectrics become highly conductive, it destabilizes the long range dipolar ordering necessary for ferroelectricity.” Thus, the statement at lines 269-271 is apparently wrong and has to be revised accordingly, avoiding mixing ferroelectric and thermoelectric features, since they indeed do not enhance each other, despite being observed in materials with the same structure or even composition.

Response: In the revised version, we changed “metallic-like conductivity happens in its paraelectric phase and the potential ferroelectricity could stabilize semiconducting feature.[145-148]” to “The heavily reduced, nonstoichiometric n-type perovskite SrTiO3−δ shows t metallic-like conductivity.[145-148]”.

From conventional views, ferroelectricity and thermoelectric features (conductivity) contradict to each other. However, it is not absolute. Long-range ferroelectricity is indeed contradictory to conducting, however, short-range ferroelectricity in the form of nanopolar regions might be possible. The pioneering idea of ferroelectric-thermoelectricity has been proposed by the experts of ferroelectrics, e.g., Clive A. Randall in the Pennsylvania State University.[Journal of the European Ceramic Society 32 (2012) 3971–3988] This idea does not aim to get a material with both ferroelectricity and thermoelectricity at the same time, especially in bulk materials. Instead, it aims to obtain a new group of thermoelectric oxides, starting from certain weak ferroelectrics like (K,Na)NbO3 or even paraelectric like SrTiO3.

Detailed comment 3: Furthermore, the previous sentence “There typical example is doped SrTiO3, which is a paraelectrics in ceramics whild could be ferroelectrics in certain strained condition.[60, 143, 144]” is full of grammatical mistakes, while the authors were asked to double check the manuscript for them. In addition, if the authors want to mention ferroelectricity induced in SrTiO3, they should also refer the recent review paper in Condens. Matter 2020, 5, 58.

Response: In the revised version, we changed There typical example is doped SrTiO3, which is a paraelectrics in ceramics whild could be ferroelectrics in certain strained condition.[60, 143, 144]” to “The typical example, the doped SrTiO3, is paraelectric in bulks while could be ferroelectric in films under certain strained condition.[60, 143, 144]”.

We added the suggested reference [Condens. Matter 2020, 5, 58] in the revised version. Please see Ref. 145.

Detailed comment 4: The meaning of sentence at line 474 “Here substitutions are doped with the same valence” is also very unclear, requiring significant revision in order to pass the message.

Response: In the revised version, we deleted it. The whole sentence is changed to “In thermoelectrics, the well-employed band structure engineering strategies to boost Seebeck coefficient, e.g., band alignment and band gap enlargement, are highly related with substitutions, like Sr doped in PbTe,[151, 152], Mg doped in PbTe,[13] and Se doped in PbTe.[10]”.

Detailed comment 4: Then, “To analysis” at line 490 should be rather “To assess”, “duo” at line 395 should be “due”, “(c1, d2)” at line 160 “(c1, c2)”, while “materials with ideally perfect crystals” at line 25 just “perfect crystalline materials”.

Response: Thanks again for carefully checking our manuscript. We have revised all according to the Reviewer’s suggestions in the revised version, please see the red marked sentences.

Detailed comment 4: Last but not least, there is still no reason to keep the word “foreseeing” in the manuscript title, since approaches are observed, confirmed, approved, but not foreseen in this manuscript.

Response: We changed the title to “Seeing structural mechanisms of optimized piezoelectric and thermoelectric bulk materials through structural defect engineering”

Round 4

Reviewer 3 Report

Now the comments are more properly addressed and the manuscript can be accepted for publication.

This manuscript is a resubmission of an earlier submission. The following is a list of the peer review reports and author responses from that submission.